# A Statistical Porosity Characterization Approach of Carbon-Fiber-Reinforced Polymer Material Using Optical Microscopy and Neural Network

**DOI:** 10.3390/ma15196540

**Published:** 2022-09-21

**Authors:** Sara Eliasson, Mathilda Karlsson Hagnell, Per Wennhage, Zuheir Barsoum

**Affiliations:** 1Scania CV AB, SE-151 87 Södertälje, Sweden; 2Centre for ECO2 Vehicle Design, SE-100 44 Stockholm, Sweden; 3Department of Engineering Mechanics, KTH Royal Institute of Technology, SE-100 44 Stockholm, Sweden; 4RISE Research Institutes of Sweden, Materials and Production, Polymers, Fibers and Composites, SE-164 40 Stockholm, Sweden

**Keywords:** Carbon-Fiber-Reinforced Polymer, porosity, Convolutional Neural Network, optical microscopy

## Abstract

The intensified pursuit for lightweight solutions in the commercial vehicle industry increases the demand for method development of more advanced lightweight materials such as Carbon-Fiber-Reinforced Composites (CFRP). The behavior of these anisotropic materials is challenging to understand and manufacturing defects could dramatically change the mechanical properties. Voids are one of the most common manufacturing defects; they can affect mechanical properties and work as initiation sites for damage. It is essential to know the micromechanical composition of the material to understand the material behavior. Void characterization is commonly conducted using optical microscopy, which is a reliable technique. In the current study, an approach based on optical microscopy, statistically characterizing a CFRP laminate with regard to porosity, is proposed. A neural network is implemented to efficiently segment micrographs and label the constituents: void, matrix, and fiber. A neural network minimizes the manual labor automating the process and shows great potential to be implemented in repetitive tasks in a design process to save time. The constituent fractions are determined and they show that constituent characterization can be performed with high accuracy for a very low number of training images. The extracted data are statistically analyzed. If significant differences are found, they can reveal and explain differences in the material behavior. The global and local void fraction show significant differences for the material used in this study and are good candidates to explain differences in material behavior.

## 1. Introduction

Lightweight design in the commercial vehicle industry is driven by demands of sustainability. The pursuit for lightweight vehicles is intensified by the reduction of fuel consumption and CO${}_{2}$ emissions, and by improving the driving range of Battery Electric Vehicles (BEVs) [1]. Investments in new technology generate an increased request for lightweight design, opening up for new, more advanced lightweight materials such as Carbon-Fiber-Reinforced Polymers (CFRP) in structural components. Investments in new materials require the development of new modeling techniques. Striving towards simulation-driven design continuously improves and develops more effective modeling methods [2]. Each part of a modeling framework needs to be optimized to reach its full potential. Machine learning is a common tool today to improve the speed and accuracy of data and trend analysis. Trained neural networks offer possibilities to quickly go through and classify large amounts of data, making them suitable for streamlining development processes.

The interest in using CFRP in structural components has increased because of its excellent mechanical properties. CFRP has a high strength-to-weight ratio, and in addition to this, high fatigue and creep resistance. The properties are fit for use in the commercial vehicle industry, where the service life of vehicles continuously increases and high-cycle fatigue is becoming a prominent topic [3]. Commercial vehicles are driven in varying environments, and CFRP’s mechanical properties are promising for handling different coupling effects. Xian et al. [4] show the potential of CFRP exposed to elevated temperatures and moisture under loading. When introducing new materials in a design process, it is essential to understand their behavior. The behavior of anisotropic materials, such as CFRP, is challenging. Numerous aspects affect the mechanical properties, e.g., fiber properties, matrix properties, layup, bonding strength, manufacturing method, and defects.

One of the most common manufacturing defects today is voids. Suppressing the formation of voids in today’s manufacturing techniques for CFRP materials comes at a high cost. For modern manufacturing techniques that target lower production costs and faster production times, the formation of voids is inevitable [5]. The paradigm, *Defect Damage Mechanics*, defined by Talreja [6], discusses the quantification of the production process to minimize cost while still fulfilling the mechanical requirements. Stamopoulos et al. [7] showed that porosity reduces all matrix-dominated material properties, and de Almeida et al. [8] concluded that propagation initiated from voids has detrimental effects on the fatigue life. Therefore, it is essential to characterize the material quality with regard to voids if it is to be introduced in a high-volume production industry.

There are many different void characterization techniques used to characterize materials. Studies have been performed comparing common techniques [9], and they all have advantages and limitations. More advanced techniques to characterize voids include ultrasonic testing and X-ray Computed Tomography (CT). Ultrasonic testing is typically used as an industry quality assurance measurement technique [10], while X-ray CT is an advanced technique that is becoming increasingly popular [11]. A method to effectively determine constituent fractions is thermogravimetric analysis [12]. The major review on voids in CFRP by Mehdikhani et al. [5] summarizes the last decades’ research on voids and discusses possible techniques for void characterization. Optical microscopy is one of the most commonly employed imaging techniques for void content evaluation, and it is commonly used in institutions and easily available and performed by an engineer. The constituent fractions are determined by defining the micrograph area fractions. The area fractions are calculated from random selections of material sections and represent the constituents’ volume fractions [13]. With optical microscopy, a statistical analysis using only 20–25 images can determine the void content [14], and it has the capability to reach an accuracy of 0.2% [15]. In image analysis, the classification of micrographs is common and continuously developed. In biomedicine, the work is growing, and microscope software such as Aivia (by Leica Microsystems, Inc) implements Artificial Intelligence (AI) solutions to analyze images. Ilastic [16] is an open-source and more simplistic software. Ilastic conducts pixel classification, and a random forest classifier is trained and applicable to a wide range of segmentation problems.

The trend of neural networks is becoming more popular in composite research. A common area is the prediction of mechanical properties [17,18,19,20]. Neural networks have also recently been introduced for the segmentation of microstructures. Ge et al. [21] reviewed the applications of deep learning on microscopic image analysis and its possibilities, and Galvez-Hernandez et al. [22] investigated interlaminar voids and dry areas in uncured prepreg based on images from micro-CT scanning. They explored the benefits of using machine learning and found that machine learning consistently exceeded a thresholding approach. Machado et al. [23] automated the void content prediction of composite laminates using a neural network overcoming issues and limitations for a thresholding approach. The studies by Galves-Hernandez et al. and Machado et al. use the well-known U-Net architecture developed by Ronneberger [24]. The U-Net is a Convolutional Neural Network (CNN) suitable for image segmentation. The neural network makes it possible to analyze many images in a short time compared to the manual labor it would typically require to recognize the constituent of a micrograph. Using a neural network enables the streamlining of modeling frameworks and the optimization of data input or output. The current study evolves the analysis of micrograph data from a previously developed framework by Eliasson et al. [25], illustrating the possibility of making a part of the framework more effective.

An accurate characterization of material constituents and porosity is essential when analyzing and modeling the mechanical properties of CFRP materials. When predicting the effects of voids, not only the void fraction is essential. Shape, size, and location of the voids are also important parameters [26,27]. Based on studies found in the literature referred to above, some studies have focused on producing test data and developing models to analyze and better understand the behavior and damage mechanisms in CFRP materials and structures. Most previous studies on material characterization lack a method to efficiently handle the extracted data. These studies are scarce about addressing a deeper statistical analysis of void content or showing the potential, development and implementation of neural networks. This motivates the current study on developing an approach for statistical porosity characterization of a CFRP material using a neural network.

In the current study optical microscopy is utilized to extract material characteristics from micrographs which are then statistically characterized in a novel and generally applicable manner. An increased amount of data is required to improve the accuracy of such a characterization, which can require a time-consuming analysis. Therefore, it is also demonstrated that by implementing a neural network the analysis of large amounts of data can be highly automated.

## 2. Methodology

### 2.1. Material

The material in the study was a Unidirectional (UD) CFRP prepreg. The UD tape has an epoxy matrix system called “snap cure”. The matrix system has a cure time of 120 s at 130 °C, suitable for high-volume production components. The composite laminate was stacked with a cross-ply layup, $\left[0_{3}^{\circ }/90_{6}^{\circ }/0_{3}^{\circ }\right]$. The material system was prepared using heated compression molding. Compression molding commonly leads to entrapped air when curing a composite material with high speed.

Three different plates were manufactured and used in the study. The manufacturing variables are presented in Table 1. Changes were made in the manufacturing process attempting to reduce voids. Two plates (Plate 1 and 2) were manufactured using the same methodology and only adjusting the compression pressure (Figure 1a). A debulking process was conducted on the third plate (Plate 3). When debulking the uncured laminate three layers of prepreg were added to the uncured stack, followed by vacuum-bagging and depressurizing. The process was repeated until the full laminate stack was reached (Figure 1b).

### 2.2. Optical Microscopy

Micrograph samples (Figure 2) looking into the fiber direction were analyzed to determine void fractions of the manufactured plates. The uncertainty of microscopy is a well-known topic and the absolute and correct void fraction can only be calculated by covering the complete surface of a sample. Furthermore, a micrograph sample only covers a small percentage of the complete component making the method section-biased. In the case of a simple plate geometry the samples were prepared from statistically representative positions of the plates; center and edge. For the simple plate geometry, it is considered enough to randomly vary the sample positions in these areas since they represent two extreme positions regarding air evacuation during the manufacturing process. However, if the geometry was more complex, it is important to remember that the micrograph samples would only reflect a local void fraction.

Two centimeters wide micrograph samples were cut using a diamond blade to minimize surface damage and then polished with up to P4000 grit silicon carbide abrasive papers to reach an undamaged surface. The micrographs were acquired with an Olympus BX53 microscope, using a magnification of 5× and 50× (Figure 2).

A micrograph sample had an area of approximately 20 mm${}^{2}$ looking into the fiber direction. The 5× magnification micrographs covered the entire surface of the sample. The micrograph images with 50× magnification were acquired consistently and systematically along the sample surface. A micrograph taken with 50× magnification covered approximately 0.05 mm${}^{2}$, meaning 400 micrographs would be needed to cover the entire surface of a micrograph sample. To determine the material’s void fraction a cross-section of at least 80% of the sample should be analyzed to have an error below 15% [28]. Therefore, the 50× magnification micrographs were not used for void fraction calculations since this would take too much time to acquire the needed images from each micrograph sample. The 50× magnification micrographs were used to extract the void data (shape, size, and location), and the 5× micrographs were used to decide the void fraction. A summary of the data available for the analysis is found in Table 2.

### 2.3. Image Analysis

The constituent fractions were determined using quantitative optical microscopy to calculate the micrograph area fractions [29]. Different image analysis techniques were used and compared to analyze the micrographs. The computation of void fraction was performed using two different methods; selection method, and thresholding, referred to as SM and TH, respectively. The value extracted by the selection method was used as the baseline. The micrographs used for the analysis of void data were first segmented with the help of a neural network and then the data was extracted.

#### 2.3.1. Selection Method (SM)

When using the selection method a raster graphics editor is favorably used to classify each pixel manually, and for constituent characterization, it was especially useful to isolate specific pixels. The voids are selected with the help of a boundary detection tool. For this study, the raster graphics editor Adobe Photoshop${}^{®}$ was used to create selections for the voids in the micrographs and determine the area fractions. The result would be precise but affected by the user’s choice and opinion [29]. The difference between users is not analyzed and is a limitation of this study.

#### 2.3.2. Thresholding (TH)

Thresholding is an established method for image segmentation and time-saving compared to the selection method. Thresholding assigns a label to each pixel based on the grayscale value of that pixel. Ranges of grayscale values are categorized, and if the grayscale value of a pixel was within a certain range, the pixel belonged to a specific category. A problem with the thresholding method was that the actual threshold needed to be set manually. Voids are always primarily black and explicitly darker than their surroundings, resulting in an easier segmentation and choice of grayscale range. The software used for thresholding was Fiji [30]. Fiji is an open-source, free image processing program recommended for this use.

The grayscale values are dependent on image and camera settings. A limitation was that scratches and other defects in the micrograph sample could have the same grayscale values as other features, which could lead to a false classification of pixels. To illustrate the difficulties with thresholding a histogram is created for each feature of a micrograph (Figure 3). Fibers do not have a distinct peak, and it is not straightforward what a threshold value for fibers would be without interfering with the matrix values.

### 2.4. A Neural Network

The manual labor needed to use the selection method and thresholding is significantly high. An automated approach that accurately assesses the required information could drastically reduce the time needed to extract the data. Deep Convolutional Neural Networks (DCNNs) are recognized as a powerful technique for image classification [31]. A Convolutional Neural Network (CNN) with a U-Net architecture [24] was used in the current study and this neural network falls under the category of supervised learning, where the ultimate objective is to map input data or an input layer to an output layer. The neural network characterized the three phases, fiber, matrix, and void, reducing the time to analyze micrograph fractions.

The U-Net exists of an encoder and a decoder, also known as a contracting path and an expansive path, that generates the U-shape. Simplifying the structure of the U-Net, the encoder extracts and recognizes features at different levels in the image, and the decoder localizes the features in the pixel space. The U-Net allows us to segment the image pixel by pixel and labels each pixel of the micrographs to its corresponding class: void, matrix, or fiber. Figure 4 shows the U-Net architecture used in the current study. The input was a 2D image (572 × 572 pixels). The contracting path consisted of repeated execution of two 3 × 3 convolutional layers followed by a Rectified Linear Unit (ReLu) activation function and one 2 × 2 max pooling operation that downsampled the feature maps by half. For each downsampling step, the number of features is doubled. The expanding path consisted of upsampling the feature map followed by a 2 × 2 up-convolution that halved the number of feature channels. The resulting feature map was concatenated with the feature map from the equivalent block from the contracting path. This step was followed by two 3 × 3 convolutions, each followed by a ReLU. At the final layer, a 1 × 1 convolution with a soft-max activation function was used to map the features to the desired number of classes, creating a probability map, one for each class, translated to one final predicted classified image. The convolutional layers are unpadded, and the output image size was 388 × 388 pixels.

#### 2.4.1. Implementing the Neural Network

The implementation of the U-Net was set up utilizing the module Keras in TensorFlow (tensorflow.keras) [32]. After careful consideration, the network settings were chosen based on input from running and training the network while monitoring the results. Only the essential functions and settings of the networks are highlighted and can be found in Table 3. In TensorFlow, the learning rate was reduced if any parameters stopped improving to control the training better throughout each epoch. The minimum learning rate used was $10^{-6}$.

#### 2.4.2. Training the Neural Network

The U-Net is known for efficiently using a small amount of input data and successfully training a model with high accuracy. The input data was varied between 50, 100, and 200 training images to compare the accuracy and the amount of training data needed. For this comparison the network was trained using only images from one of the plates (Plate 1) to see how successfully it could predict unseen material (Plate 2 and Plate 3). The comparison between the number of training images was limited to 100 epochs. In a second step the process was optimized to train the best model. Two optimized models were trained, one using the 200 training images from Plate 1, but running it for 500 epochs. The second had optimized input data, adding training images from Plate 2 and Plate 3, and running for 200 epochs. The trained networks are summarized in Table 4. The final results were extracted with a script loading the trained U-Net and making a prediction for an entire validation micrograph.

The ground truths were prepared using the selection method. The ground truths pixel values ranged from zero to two, and each number represents one class; void, matrix, and fiber, respectively (Table 5). Training the U-Net, the weights of each feature must be similar. The micrographs were cropped to the correct size for the network (572 × 572 pixels) and weighted. The weights of the training images are presented in Table 6.

### 2.5. Extraction of Void Data

The shape, size, and location of voids are extracted from segmented micrographs predicted with the trained neural network TI300E200. A script was run where a region of pixels represented each void in the micrograph. Tiny voids were considered noise and the limit for a region to be considered as a void was 20 μm${}^{2}$. This area corresponded to a void slightly smaller than a fiber. The void area was based on the number of pixels of the region, and a transformation coefficient was calculated and extracted from the micrographs.

The void shape was fitted to a circle and an ellipse (Figure 5), with coinciding centers located in the Center of Gravity (CoG) of the void region. The elliptical shape was fitted based on an assumption of equal normalized second central moments of the void region and the ellipse. The ellipse shape was described with area, and major and minor axes. The circle was fitted based on the assumption that the circle had the same area as the void region. The circle was described with area and radius. To find the location and distribution of the voids, the first, second, and third Nearest Neighbors (NN) were extracted.

### 2.6. Statistical Analysis

Data extracted from the micrographs were analyzed with different statistical methods. It is essential to get an overview of the extracted data and to understand and conclude differences between the plates. The statistical analysis was performed using MATLAB [33]. For the statistical analysis a significance level of α=0.5 was used, and the probability was set to 95 %. The void fraction and void data for all three plates were analyzed.

Differences between the manufactured plates were analyzed, and comparing two different population means, a t-test was used. However, a t-test assumes normally distributed data, which must be checked. The Shapiro–Wilk test [34] was used to check if the data were normally distributed. The non-parametrical option for a t-test is the Mann–Whitney U-test (also known as the Wilcoxon test). This test is essentially the same as the t-test but for independent samples. The null hypothesis will state no difference between the two population means analyzed.

When comparing differences between the plates regarding the extracted void data, there are more than two groups (three plates), which means that the Analysis of Variance (ANOVA) [35] was preferably used. The one-way ANOVA is a parametrical method and assumes that the data are normally distributed. However, the ANOVA is robust and is not sensitive to moderate deviations. The ANOVA is performed with this in mind, and results are carefully analyzed to avoid a false-positive result. The null hypothesis stated that there was no significant difference between the means of the groups. If the null hypothesis was rejected, a post hoc test was run using LSD and Bonferroni. The LSD is commonly used, while the Bonferroni complements it by being more conservative.

## 3. Results

### 3.1. The Neural Network

The amount of data and the number of EPOCHS highly affect the run time. During the training of the U-Net, both the loss and validation data were monitored. The training data for each U-Net trained in TensorFlow using 200, 100, or 50 training images are presented in Figure 6a. The optimized models are shown in Figure 6b.

The Validation Images (VI) used to evaluate the network, VI1 (Plate 2) and VI2 (Plate 3), are presented in Figure 7. The validation images were chosen to represent different specifics, e.g., strangely shaped voids, bright fibers, scratches, and blurry focus. When predicting an entire micrograph, the edges were slightly problematic due to the bad resolution of the micrograph edges. Therefore, the final predicted entire micrograph was centrally cropped, maintaining 95% of the micrograph.

#### Neural Network Performance

An illustration of the wrong classified pixels for VI1 is presented in Figure 8. The wrong classified pixels are marked in yellow color, and it can be seen they are all found in the boundary between two constituents, i.e., around fibers or around voids. The trend was the same for VI2. To see which feature the network has the most difficulty predicting, a matrix is presented summarizing the wrong classified pixels, and their true value compared to the predicted value (Table 7 and Table 8). Fibers are mainly predicted as matrix; for VI2 this is almost 80% of the wrong classified pixels. For VI1, it is more evenly divided between fiber predicted wrong as matrix, or the other way around.

Comparing predicted fractions for each constituent to the ground truth of the validation images (Figure 9), the optimized networks perform the best and the void prediction is good, with a maximum fault of approximately 7%. The prediction of fiber and matrix has a higher percentual deviation from the true fractions. The void fraction is very well predicted using only 100 training images supporting the strength of the U-Net. The improvement between the TI200E100 and the optimized networks is marginal for the fiber and matrix prediction.

Evaluating the void fraction and comparing predictions between the three different methods, SM, TH, and U-Net, there was minimal variation (Figure 10). The results support the accuracy of the different methods generating similar results.

### 3.2. Statistical Analysis

#### 3.2.1. Void Fraction Data

Basic descriptives and basic statistical tools were used to understand the void data. The mean, median, standard deviation, variance, max, and min values for data extracted from 5× magnification micrographs using the selection method and thresholding are given in Table 9 and Table 10, respectively. It can be seen that the mean value of the void fraction for the three plates decreases, with the largest value being for Plate 1. The mean value for Plate 1, differs the most between the two methods. This difference could be due to the quality of the micrographs and Plate 1 not being as suited for thresholding.

To test for normality of the distribution, the Shapiro–Wilk test was used. The null hypothesis was rejected for all data sets. Details are presented in Appendix B.

The Mann–Whitney U-test was used to compare the selection method and thresholding. The null hypothesis stated that the void fraction was identical for the two methods. The results are presented in Table 11. The null hypothesis was rejected for Plate 1 and there was a significant difference between the methods. This result supports that the quality of the micrographs from Plate 1 was not suitable for thresholding, and moving forward in the analysis, the selection method data was used.

The mean void fraction differed but this did not ensure a significant difference between the plates’ void fractions. An ANOVA was used to analyze the difference between the plates. The null hypothesis stated that there was no difference between the void fraction of the plates. The ANOVA results are presented in Table 12, where the null hypothesis was rejected, and there was a significant difference between the plates. To determine which plates differ from each other a post hoc test was conducted.

The results for the post hoc test are presented in Table 13. For both post hoc tests, there was no significant difference between Plate 2 and Plate 3 (*p*-value > α). Plate 1 differs from Plate 2 and Plate 3, telling us that the void volume fraction was significantly higher for Plate 1. The final results for the mean void fraction of the plates are summarized together with the Confidence Interval (CI) in Table 14 and Figure 11.

#### 3.2.2. Void Data

The 50× magnification micrographs run through the trained network generated information for the statistical analysis of void data. The void data covered information on void shape, size, and location. The total amount of analyzed voids, extracted with the U-Net, is summarized in Table 15. The average number of voids per image is also presented.

A summary of the void shape parameters for each plate is presented in Table 16, Table 17 and Table 18. Histograms are presented for the different variables to get an overview of the extracted data (Figure 12 and Figure 13). The first, second, and third NNs were also calculated and plotted in histograms (Figure 14).

Not only void fraction could affect a material behavior, but also the different void data parameters that were extracted. The void data parameters were compared using ANOVA and the analyses are summarized in Table 19 with their respective null hypothesis and result.

For the analyses where the ANOVA null hypothesis was rejected, a post hoc test was conducted to identify which plates differ from each other. A summarized result is found in Table 20. It is concluded that there is a significant difference between the third NN distance for Plate 2 and Plate 3. Meaning that the third nearest neighbor is closer for Plate 3. The number of voids in a micrograph has a significant difference for Plate 1 and Plate 2, and, Plate 2 and Plate 3. Concluding that there are more voids in the micrographs for Plate 2.

## 4. Discussion

### 4.1. The Neural Network

Using a neural network the approach for porosity characterization becomes highly automated and can be used to analyze large amounts of data with high efficiency. However, it can be a big step to introduce a neural network to a currently manual process.

#### Neural Network Performance

The network successfully identified the features: void, matrix, and fiber. The wrong classified pixels are mainly found in the boundary between two features (Figure 8c) and the majority of boundary pixels are between fiber and matrix. Recall that the fiber and matrix phase overlapped in grayscale color, which made them harder to label (Figure 3). However, the network seems to understand and distinguish well between fiber and matrix, it does not only look at pixel values. The network is robust and considers variables such as shape, size, and placement. The study is taken one step further compared to Machado et al. [23], who showed the possibilities of identifying voids. In this study, all micrograph constituents were identified and statistically characterized. This points towards further possibilities of the approach to identify, e.g., fiber tows, and layers, and then calculate or analyze orientations and deviations. The approach could also be used on CT-scanned micrographs as in [22].

Increasing the number of training images and epochs directly correlates to lowering the amount of wrong classified pixels (Figure 9). Depending on the network’s intended use, you can choose a suitable level of detail. The run time for training the network increases with a need for higher accuracy, and the number of training images can be limited to identify voids. The network is very good at finding variations and details, and the error for void prediction is below 10% using only 100 training images and 100 epochs.

Comparing the results from the neural network to the selection method and thresholding, they all get similar results. This supports the accuracy of the approach, however, there is still the question of section-bias for optical microscopy. Therefore, it is important to know that the more variation and random data you use, the more accurate the results will be. For a complex geometry it is important to understand what you are analyzing and if the local void fraction for samples would be representable for the whole part. The use of the neural network allows for analyzing more data more efficiently making the results more reliable. All methods are an approximation and even if you could scan an entire part, there is still an insecurity in determining what is a void and what is not.

### 4.2. Statistical Analysis

It is concluded that the void fraction is significantly higher for Plate 1 and there is no significant difference between the other plates. Statistically characterizing the data can contribute to a better understanding of the material. Even though it is tempting to only look at a mean value, the statistical analysis can reveal another truth. The extracted features on void shape and size have not yet been linked to composite damage, and the significant differences found or not found are not related to a true material behavior, which is a limitation of this study. The shape and size of voids did not show any significant differences between the three plates, however, it is known that void shape and size do affect the material properties and could initiate damage [26,27]. There was a significant difference between distance to the third NN and number of voids in a micrograph. Which are also important parameters since this connects to a varying local void fraction. The local void fraction is also important to consider when modeling the material behavior [25].

## 5. Conclusions

The work presents a statistical porosity characterization approach for a UD CFRP material. Micrographs have been successfully segmented, classifying the constituents: voids, matrix, and fibers, with the help of a neural network. Conclusions from the study are summarized below:The findings in this work can help streamline other repetitive manual processes that require large amounts of data making them more efficient. To open up for the possibility of implementing a neural network, the authors have decided to include and publish the script for the implementation of the neural network which could serve as a guideline for the research community. See Appendix A.The study shows that with only a few training images certain constituents in a micrograph can be identified with high accuracy. For the current material and micrographs the neural network needed only 100 training images to identify voids and void fractions with an accuracy above 90%.The neural network performs equally to other methods used for image segmentation of micrographs and can be trained to perform even better and focus on more advanced issues to be solved.The statistical conclusions regarding the porosity of the material can help explain mechanical behavior in testing and how to use the material characteristics in a simulation approach.The global and local void fraction are good candidates to explain differences in material behavior.

## Figures and Tables

**Figure 1 materials-15-06540-f001:**
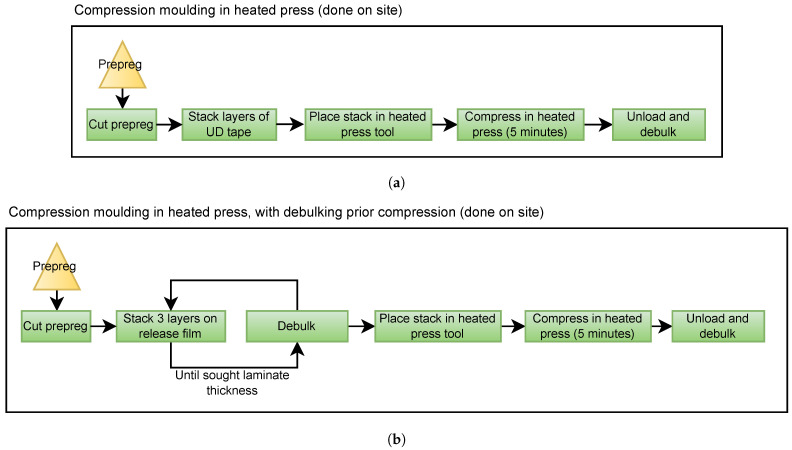
The flow of the different compression molding manufacturing processes used, (**a**) without debulking, and (**b**) with debulking.

**Figure 2 materials-15-06540-f002:**
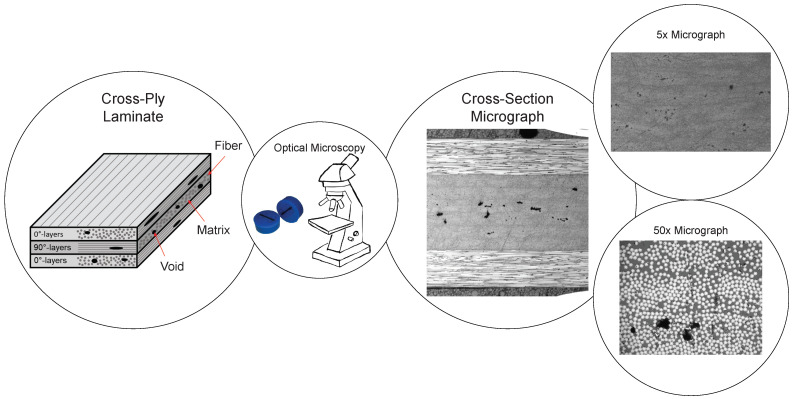
An illustration of the optical microscopy work flow showing the cross-ply laminate and the micrographs used for the analysis.

**Figure 3 materials-15-06540-f003:**
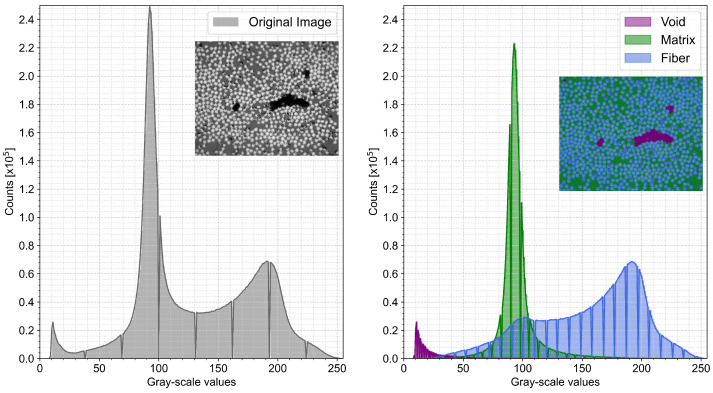
Histogram for a micrograph and each of the constituents; voids, matrix, and fibers.

**Figure 4 materials-15-06540-f004:**
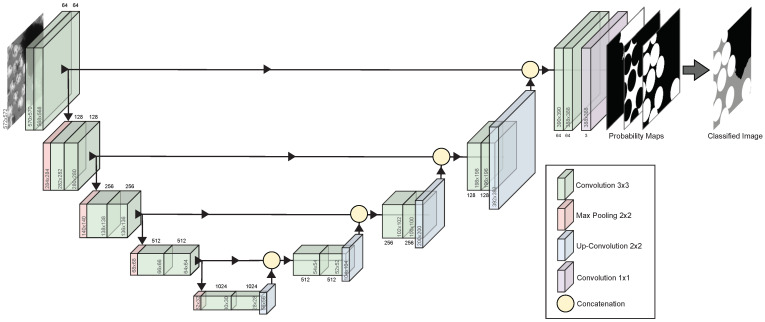
A schematic image of the U-Net architecture.

**Figure 5 materials-15-06540-f005:**
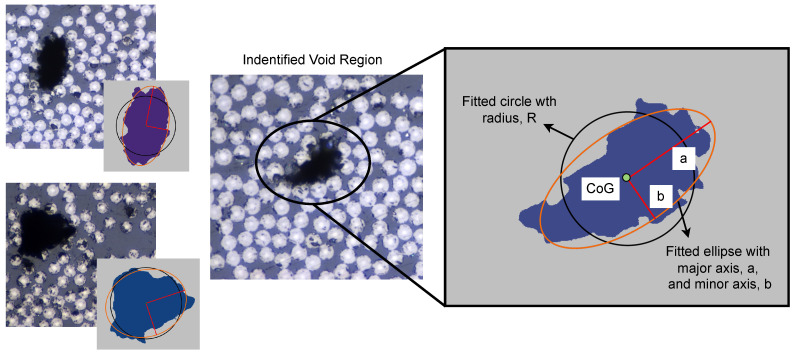
Identified void regions with the void fitted to a circular and an elliptical shape.

**Figure 6 materials-15-06540-f006:**
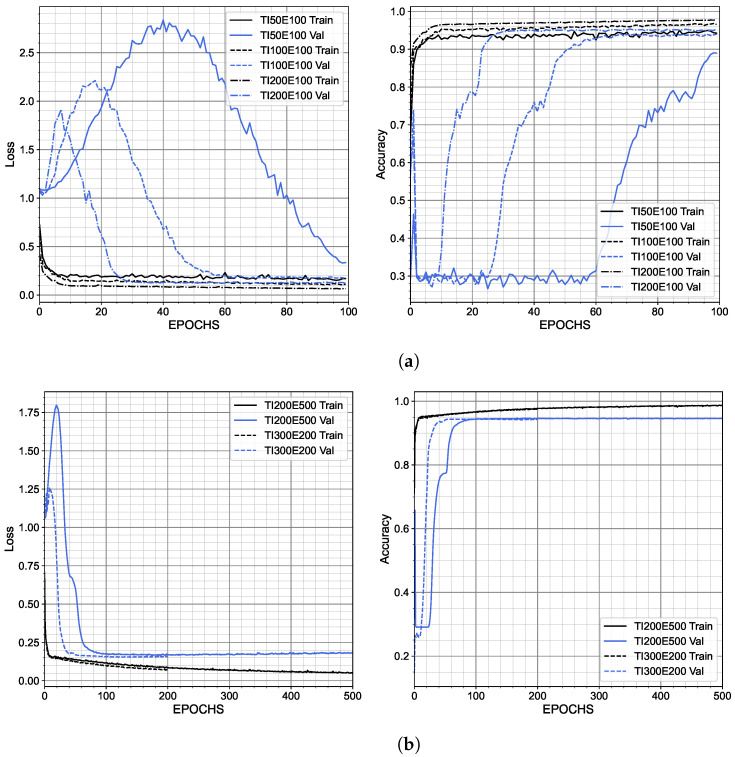
Accuracy and loss for the training of the U-Net, (**a**) for different numbers of training images, and (**b**) for optimized models.

**Figure 7 materials-15-06540-f007:**
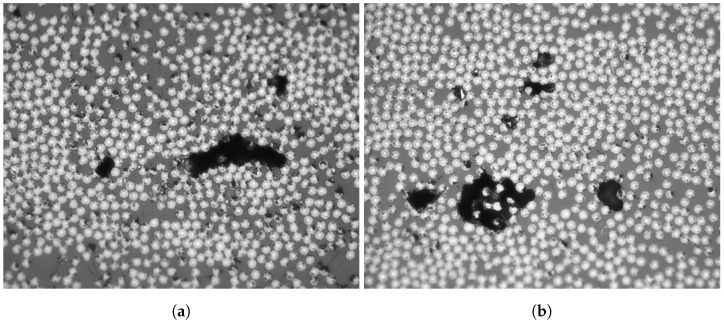
The validation images for evaluation of the trained network (**a**) VI1 from Plate 2, and (**b**) VI2 from Plate 3.

**Figure 8 materials-15-06540-f008:**
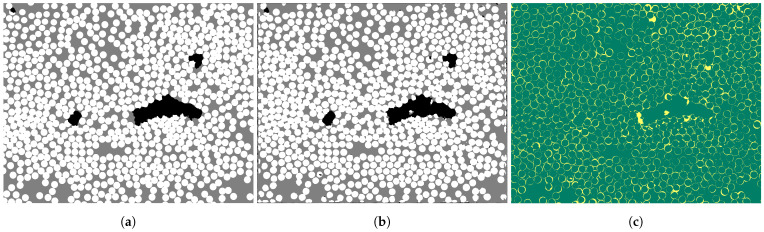
VI1 (**a**) ground truth, (**b**) predicted segmented micrograph, and (**c**) wrong and correct predicted pixels in yellow and green, respectively.

**Figure 9 materials-15-06540-f009:**
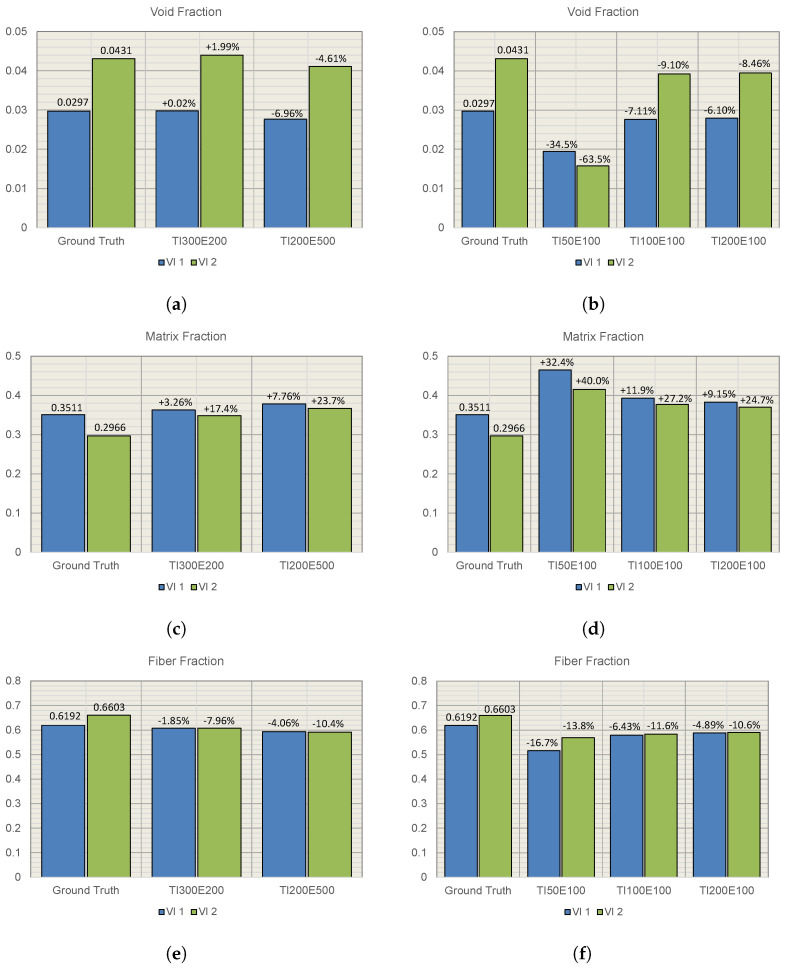
Comparison of constituent fractions, predicted for VI1 and VI2, and ground truth, (**a**,**b**) void fraction, (**c**,**d**) matrix fraction, and (**e**,**f**) fiber fraction.

**Figure 10 materials-15-06540-f010:**
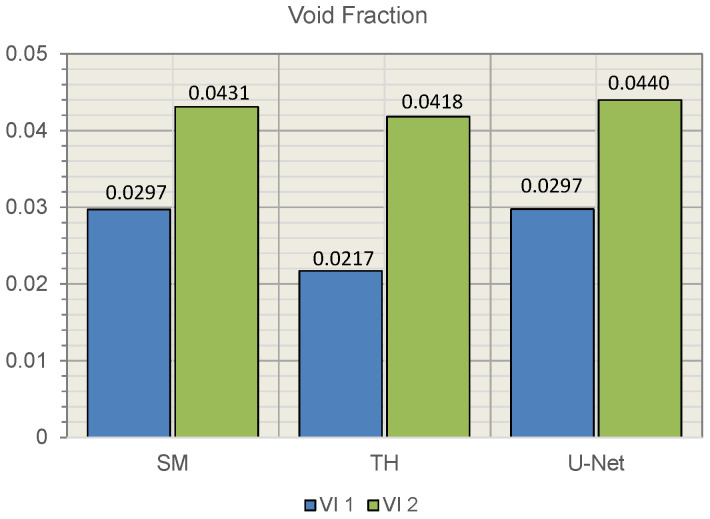
Comparing calculated void fraction between different methods.

**Figure 11 materials-15-06540-f011:**
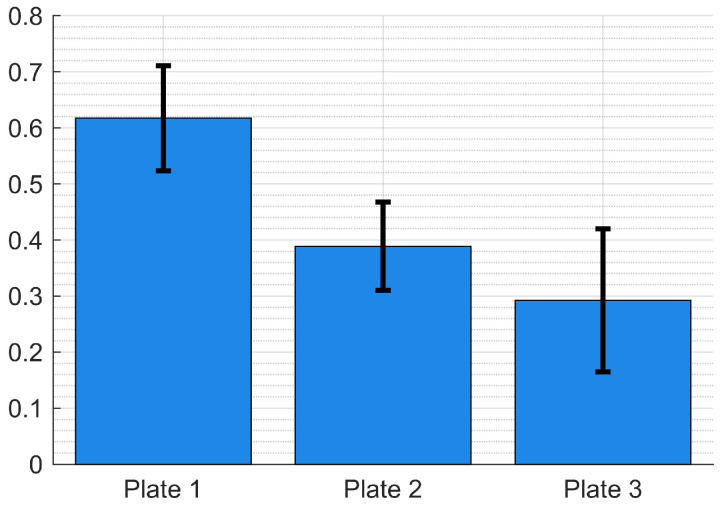
Plot illustrating the mean void fraction for the plates and their resp. 95% CI.

**Figure 12 materials-15-06540-f012:**
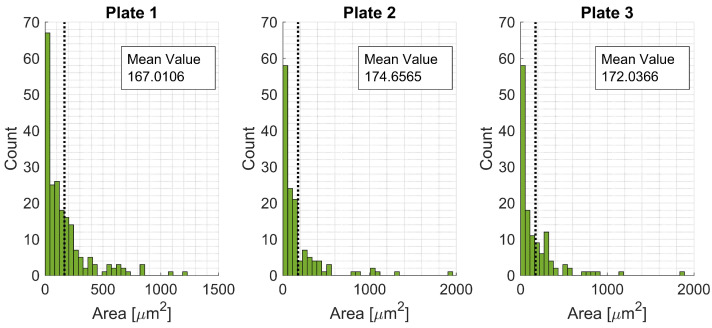
Histograms for void areas for Plates 1, 2, and 3.

**Figure 13 materials-15-06540-f013:**
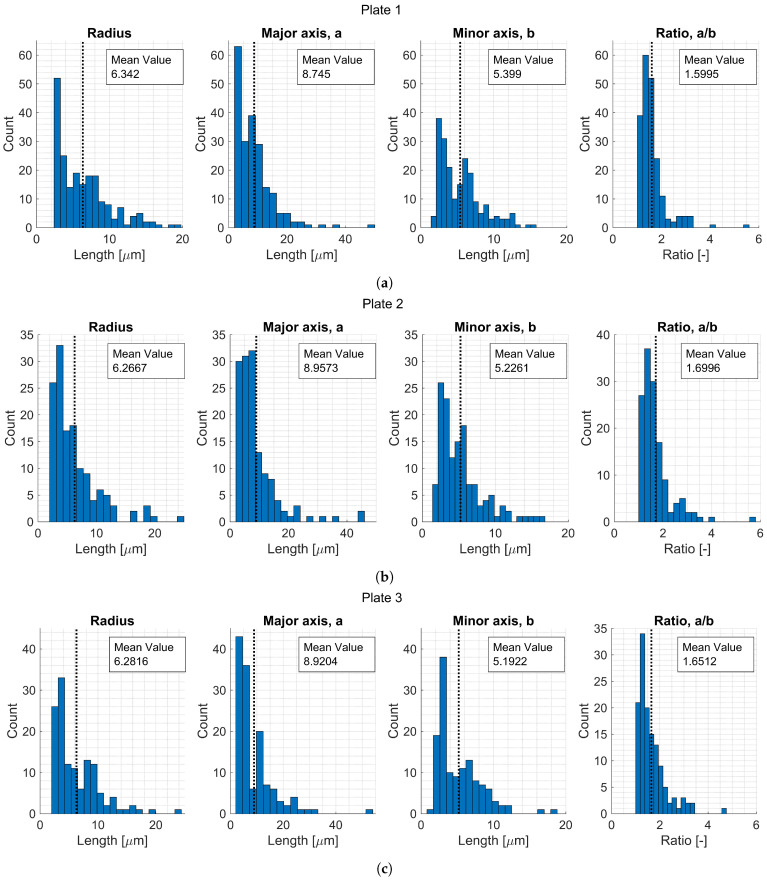
Histograms for void data variables for (**a**) Plate 1, (**b**) Plate 2, and (**c**) Plate 3.

**Figure 14 materials-15-06540-f014:**
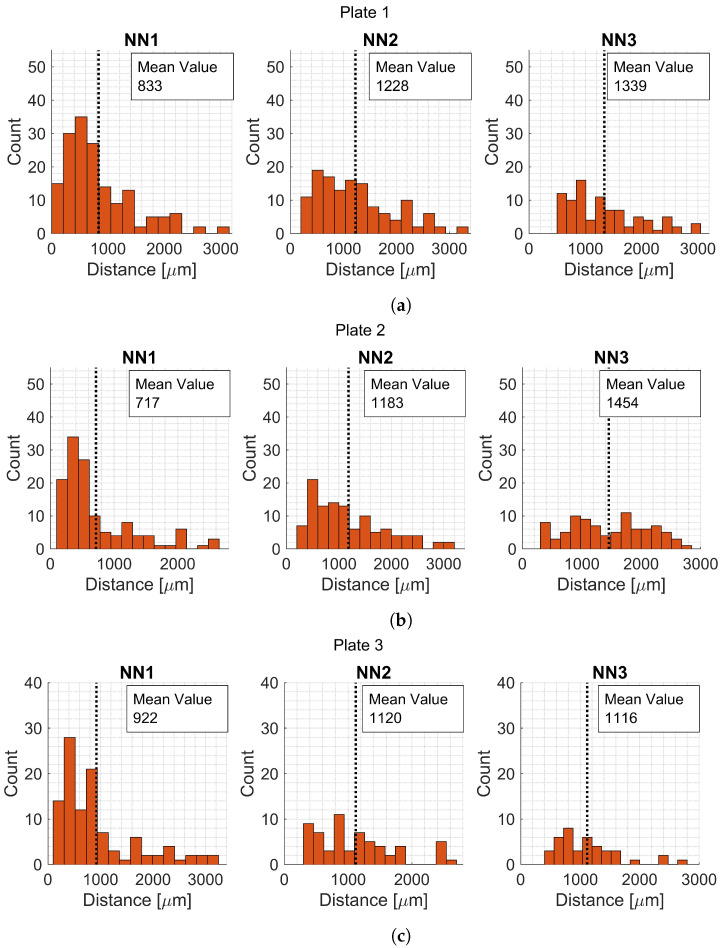
Histograms for first, second and third NN for (**a**) Plate 1, (**b**) Plate 2, and (**c**) Plate 3.

**Table 1 materials-15-06540-t001:** Processing parameters for the three manufactured plates, presented in rounded numbers.

Plate	No. Plies	Size [cm]	P [MPa]/F [kN]
1	12	34.5 × 34.5	0.7/86
2	12	31 × 31	1/100
3	12	33 × 33	0.9/100

**Table 2 materials-15-06540-t002:** Sample matrix showing the number of micrographs available for the magnifications used.

Magnification	Plate 1	Plate 2	Plate 3	Total No. Samples
5×	64	64	38	166
50×	177	80	127	384

**Table 3 materials-15-06540-t003:** Training settings for TensorFlow.

Setting	Value
Input image size	572 × 572
Output image size	388 × 388
Number of classes	3
Batch size	8/20
Buffer size	1000
Learning rate	$10^{-4}$
Optimizer	ADAMS
Loss	Categorical Cross-Entropy

**Table 4 materials-15-06540-t004:** Summary of trained networks.

Name	Training Images	EPOCHS
TI50E100	50	100
TI100E100	100	100
TI200E100	200	100
TI200E500	200	500
TI300E200	300	200

**Table 5 materials-15-06540-t005:** Pixel values for each feature in the extracted ground truths.

Pixel Value	Feature	Colour
0	Void	Black
1	Matrix	Grey
2	Fiber	White

**Table 6 materials-15-06540-t006:** The weights of each constituent in the different training images.

No. Training Images	Feature	Weight
	Void	1.729
50	Matrix	1.096
	Fiber	0.663
	Void	1.831
100	Matrix	1.067
	Fiber	0.659
	Void	2.426
200	Matrix	1.033
	Fiber	0.617
	Void	1.717
300	Matrix	1.114
	Fiber	0.658

**Table 7 materials-15-06540-t007:** Wrong Prediction Matrix for VI1.

	Predicted	0	1	2
True				
**0**		na	0.5%	1.05%
**1**		0.88%	na	41.95%
**2**		0.68%	54.94%	na

**Table 8 materials-15-06540-t008:** Wrong Prediction Matrix for VI2.

	Predicted	0	1	2
True				
**0**		na	2.37%	2.07%
**1**		1.12%	na	9.95%
**2**		4.51%	79.98%	na

**Table 9 materials-15-06540-t009:** Data extracted for the plates with SM.

Plate	Mean	Median	Std. Dev	Variance	Max	Min
Plate 1	0.617	0.531	0.376	0.141	1.705	0.119
Plate 2	0.389	0.259	0.314	0.099	1.279	0.009
Plate 3	0.292	0.051	0.388	0.151	1.186	0.0
All	0.455	0.360	0.379	0.143	1.705	0.0

**Table 10 materials-15-06540-t010:** Data extracted for the plates with TH.

Plate	Mean	Median	Std. Dev	Variance	Max	Min
Plate 1	0.444	0.420	0.245	0.060	1.110	0.070
Plate 2	0.312	0.255	0.204	0.042	0.942	0.027
Plate 3	0.268	0.110	0.308	0.095	1.212	0.010
All	0.353	0.293	0.256	0.066	1.212	0.010

**Table 11 materials-15-06540-t011:** Results for the Mann–Whitney U-test.

Dataset	*p*-Value	Conclusion
Plate 1 SM vs. TH	0.00959	$H_{0}$ is rejected
Plate 2 SM vs. TH	0.53399	$H_{0}$ is not rejected
Plate 3 SM vs. TH	0.16701	$H_{0}$ is not rejected

**Table 12 materials-15-06540-t012:** Results for the ANOVA comparing the void volume fractions for the three plates.

Data	Sum of Squares	df	Mean Square	F	*p*-Value
Between Groups	2.974	2	1.487	11.718	<0.001
Within Groups	20.686	163	0.127		
Total	23.660	165			

**Table 13 materials-15-06540-t013:** Results for the post hoc test were conducted after the ANOVA, analyzing the void volume fraction of the three plates.

Post Hoc Test	Plate	Plate	*p*-Value	95% CI	
				Lower Bound	Upper Bound
LSD	Plate 1	Plate 2	<0.001	0.10435	0.35305
		Plate 3	<0.001	0.18095	0.46907
	Plate 2	Plate 1	<0.001	−0.35305	−0.10435
		Plate 3	0.189	−0.04775	0.24037
	Plate 3	Plate 1	<0.001	−0.46907	−0.18095
		Plate 2	0.189	−0.24037	0.04775
Bonferroni	Plate 1	Plate 2	0.001	0.07637	0.38103
		Plate 3	<0.001	0.14854	0.50149
	Plate 2	Plate 1	0.001	−0.38103	−0.07637
		Plate 3	0.566	−0.08017	0.27278
	Plate 3	Plate 1	<0.001	−0.50149	−0.14854
		Plate 2	0.566	−0.27278	0.08017

**Table 14 materials-15-06540-t014:** Resulting void fraction for the three manufactured plates with their resp. 95% CI.

Dataset	Mean	CI Lower Bound	CI Upper Bound
Plate 1	0.6174	0.5236	0.7112
Plate 2	0.3887	0.3102	0.4672
Plate 3	0.2924	0.1648	0.4200

**Table 15 materials-15-06540-t015:** Void data information for the three plates.

Plate	Total No. Voids	Mean No. Voids per Image
Plate 1	205	1.3423
Plate 2	138	2.5849
Plate 3	131	1.0480

**Table 16 materials-15-06540-t016:** Void data for Plate 1.

Variable	Mean	Median	Std. Dev	Variance	Max	Min
Area [μm${}^{2}$]	167.01	97.987	199.52	39810	1215.1	20.179
Radius [μm]	6.3417	5.5848	3.6065	13.007	19.667	2.5344
Major Axis, a [μm]	8.745	7.5369	6.2215	38.707	48.695	2.8456
Minor Axis, b [μm]	5.3986	4.9228	2.878	8.283	15.601	1.511
Ratio, a/b [-]	1.5995	1.4671	0.55665	0.30986	5.5331	1.0093

**Table 17 materials-15-06540-t017:** Void data for Plate 2.

Variable	Mean	Median	Std. Dev	Variance	Max	Min
Area [μm${}^{2}$]	174.66	79.496	271.36	73638	1939.1	20.076
Radius [μm]	6.2667	5.0301	4.0549	16.442	24.844	2.5279
Major Axis, a [μm]	8.9573	6.8423	7.1545	51.187	45.368	2.7646
Minor Axis, b [μm]	5.2261	4.5227	3.0409	9.247	16.684	1.7608
Ratio, a/b [-]	1.6996	1.539	0.64981	0.42225	5.6753	1.0177

**Table 18 materials-15-06540-t018:** Void data for Plate 3.

Variable	Mean	Median	Std. Dev	Variance	Max	Min
Area [μm${}^{2}$]	172.04	73.086	245.7	60371	1857.7	20.528
Radius [μm]	6.2816	4.8233	3.9269	15.421	24.317	2.5562
Major Axis, a [μm]	8.9204	6.2931	7.2679	52.822	53.629	2.7471
Minor Axis, b [μm]	5.1922	4.1764	2.9803	8.8823	18.617	1.5814
Ratio, a/b [-]	1.6512	1.4596	0.58867	0.34653	4.7522	1.0344

**Table 19 materials-15-06540-t019:** Table for ANOVA analyzes.

Analysis	Null Hypothesis	*p*-Value	Result	Conclusion
Comparison of void size between the three plates.	H${}_{0}$ states that there is no difference in void size between the three plates	0.9543	H${}_{0}$ is not rejected	There is no significant difference between the void size of the plates.
Comparison of distance to the first NN between the three plates.	H${}_{0}$ states that there is no difference in distance to first NN between the three plates	0.053	H${}_{0}$ is not rejected	There is no significant difference between the distance to the first NN between the plates.
Comparison of distance to the second NN between the three plates.	H${}_{0}$ states that there is no difference in distance to second NN between the three plates	0.597	H${}_{0}$ is not rejected	There is no significant difference between the distance to the second NN between the plates.
Comparison of distance to the third NN between the three plates.	H_0_ states that there is no difference in distance to third NN between the three plates	0.0202	H_0_ is rejected	There is a significant difference between the distance to the third NN between the plates.
Comparison of the number of voids in one micrograph between the three plates.	H_0_ states that there is no difference in the number of voids in one micrograph between the three plates	<0.001	H_0_ is rejected	There is a significant difference between the number of voids in one micrograph between the plates.

**Table 20 materials-15-06540-t020:** Void data for Plate 3.

Analysis	Plates		Bonferroni	LSD
Comparison of distance to third NN	Plate 1	Plate 2	0.66796	0.22265
	Plate 1	Plate 3	0.19658	0.065526
	Plate 2	Plate 3	0.015921	0.0053069
Comparison of No. voids in a micrograph	Plate 1	Plate 2	<0.001	<0.001
	Plate 1	Plate 3	0.6009	0.2003
	Plate 2	Plate 3	<0.001	<0.001

## Data Availability

Not applicable.

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
