# Peer review of "A Statistical Porosity Characterization Approach of Carbon-Fiber-Reinforced Polymer Material Using Optical Microscopy and Neural Network"

_materials, 2022, doi:10.3390/ma15196540_

Round 1

Reviewer 1 Report

The paper is well presented and may prove to be very useful for the community. Besides that I have few minor comments:

1. Many typos are there.

2. Did the authors consider the effect of voids? Explain in brief.

3. English grammar needs to be improved.

4. Figures 12 and 13 quality needs to be improved.

5. Cite this paper on carbon and epoxy composite system (for the lightweight materials application) in the Introduction section:

2020. Mechanical, microstructural, and thermal characterization insights of pyrolyzed carbon black from waste tires reinforced epoxy nanocomposites for coating application. Polymer Composites41(1), pp.338-349.

Reviewer 2 Report

An innovative research work was carried out on a statistical porosity characterization of CFRP using optical microscopy and neural network. This work has a detailed and systematic design, and some results are key to predict the internal void of CFRP plate. Before the paper is accepted, it is suggested to consider the comments below to further improve the quality of the paper. 

1# Abstract, the current way of writing does not convey some key information. Some qualitative and quantitative results and conclusions are suggested to be added to the abstract to provide the authors with latest findings.

2# Introduction, it is essential to introduce the properties and advantages of CFRP when mentioning the engineering application of materials. In addition to light weight, CFRP has excellent mechanical properties, fatigue and creep resistance, and durability. These advantages also determine the development potential of CFRP when it is used for vehicles parts when facing the complex coupling effect of environment and loading. During the long-term service process, the coupling effect may cause the performance degradation of CFRP, such as the matrix cracking, interface debonding, defects formation and crack propagation and so on. Therefore, it is suggested to summary the short/long-term performance and advantages of CFRP when exposed in complex coupling effect. Please review some recent work below, such as Composite Structures, 2021, 256: 113058. Composite Structures, 2022, 293, 115719.

3# For the void characterization techniques, in addition to optical microscopy and X-ray CT and ultrasonic testing, thermogravimetric analysis (International Journal of Fatigue, 2020, 134: 105480, Composite Structures, 2022. 281: 115060) is proved to be a very simple and effective experimental method to obtain the internal porosity of CFRP. It is suggested that the authors supplement this characterization technique.

4# In the last paragraph of the introduction, it is suggested to further emphasize the innovation of this research. And further suggest that the key problems to be solved in this paper.

5# In part 2.2, How to ensure that the selected position is more representative when the optical microscope is used to select the position to determine void fractions? As known, the distribution of internal defects for FRP is random due to complex and multifactorial effects during the sample preparation and service.

6# In the second part, the authors have obtained some testing, simulation and analysis methods of porosity. However, how do the porosity affect the mechanical properties of CFRP plates? Why not conduct mechanical property tests in this paper? It is more meaningful to study the relationship between properties and porosity.

7# The authors have conducted detailed data analysis in the results and discussions on void. There is only one problem worthy of attention, that is, how to ensure the accuracy of this model?

8# The conclusion is suggested to be written separately, including 3-4 most important information points. 

Reviewer 3 Report

- The abstract does not communicate well the main findings of the study. Specific information that could be of interest to the readers should be provided.

- The topic was presented well. But it misses highlighting the importance of the topic in the introduction and the reason for conducting the research and the novelty.

- The discussion of the results should be significantly improved by comparing the results with other more related studies

-I prefer to separate the conclusions from the discussion.

In the conclusions section, one or more conclusions derived from the study should be included. Authors should avoid including information corresponding to results or discussion.

Round 2

Reviewer 2 Report

It is suggested to accept the paper.